# Bridging Classical and Learned Priors: A Hybrid Framework for Medical Image Enhancement

**Peeyush Kumar Singh**                                    S24008@STUDENTS.IITMANDI.AC.IN
**Sneha Singh**                                                     SNEHA@IITMANDI.AC.IN
*Indian Institute of Technology Mandi, India*

**Editors:** Accepted for publication at MIDL 2026

## Abstract

Medical image enhancement faces a fundamental trade-off: classical methods preserve anatomical fidelity but over-smooth fine structures, while deep learning approaches risk generating unrealistic artifacts on limited clinical data. We introduce a hybrid framework combining classical preprocessing with pretrained diffusion priors for high-quality enhancement across modalities. Our method leverages pretrained Stable Diffusion model without requiring domain specific training. During inference, classical enhancement methods generate pseudo-labels. The frozen diffusion model leverages its learned priors to refine fine structures while gradient-based guidance anchors generation to the pseudo-label, preventing hallucinations. We demonstrate efficacy in ultrasound and MRI segmentation and achieve significant improvements in multi-class cardiac structure segmentation compared to baseline models. Critical insights include: pseudo-labels outperform multi-stage classical pipelines by providing differentiable guidance targets for diffusion models, testing segmentation models on enhanced images yields additional performance gains, pseudo-label guidance strength requires domain specific tuning to balance classical robustness with learned refinement. With extensive evaluation across imaging modalities, we show that pretrained diffusion models can enhance medical images while preserving the interpretability and diagnostic fidelity essential for clinical deployment.

**Keywords:** Medical Image Enhancement, Diffusion models, Prior-guided generation, Ultrasound, MRI, Synthesis

## 1. Introduction

Medical image enhancement is critical for improving diagnostic accuracy and enabling reliable automated analysis. However, enhancement methods face a fundamental challenge: they must improve image quality while preserving anatomical fidelity and avoiding the introduction of misleading artifacts that could compromise clinical decision-making. This is very challenging in medical imaging modalities such as ultrasound and MRI, where the presence of inherent noise characteristics such as speckle in ultrasound and Rician in MRI, may have an impact on diagnostic quality. Classical enhancement approaches have dominated medical imaging due to their interpretability. Speckle Reducing Anisotropic Diffusion (SRAD) (Yu and Acton, 2002) for ultrasound uses diffusion-based denoising for a multiplicative noise model, and Contrast Limited Adaptive Histogram Equalization (CLAHE) (Zuiderveld, 1994) enhances local contrast. Adaptive histogram equalization (SM, 1977) approaches for MRI improve tissue contrast while trying to keep brightness relationships that are important for radiological interpretation. Nonetheless, these methodologies are

inherently constrained by their dependence on handcrafted priors: they often excessively smooth intricate anatomical structures, encounter difficulties with significant degradation, and fail to adapt to varied clinical contexts without comprehensive parameter adjustment. Deep learning techniques have surfaced as formidable alternatives, with convolutional neural networks (Zhang et al., 2017; Ker et al., 2017), generative adversarial networks (You et al., 2019; Armanious et al., 2020), and more recently, diffusion models (Ho et al., 2020; Saharia et al., 2022) showcasing remarkable enhancement capabilities. Diffusion models have demonstrated exceptional efficacy in image restoration by effectively learning intricate data distributions (Song et al., 2020).

Recent works have adapted diffusion models for medical image enhancement (Song et al., 2021; Wolleb et al., 2022), including approaches that train domain specific models on paired or unpaired data. However, these supervised methods face critical limitations in medical imaging: the scarcity of ground-truth clean images, the risk of generating realistic-looking but anatomically incorrect hallucinations when training data is limited, and the computational cost of training separate models for each modality. These challenges have hindered the clinical translation of diffusion-based enhancement despite their technical promise.

We introduce a hybrid framework that synergistically combines classical methods with the refinement capabilities of pretrained diffusion models to address the challenge of enhancing degraded medical images without domain-specific training while preventing anatomical hallucinations which is a critical requirement for clinical adoption. Our approach uses classical preprocessing to generate pseudo-labels that serve as differentiable guidance targets during diffusion sampling. Inspired by underwater image enhancement diffusion priors (Du et al., 2025), we apply gradient-based constraints so the frozen Stable Diffusion model (Rombach et al., 2022) refines structures beyond classical limits while remaining anchored to targets, preventing hallucinations. We validate our framework on ultrasound cardiac segmentation using the CAMUS dataset (Leclerc et al., 2019) and MRI cardiac segmentation using ACDC dataset (Bernard et al., 2018), demonstrating that pretrained diffusion models can enhance medical images when properly constrained by domain-appropriate classical priors.

**Our main contributions are:**

- A training-free framework combining classical preprocessing with pretrained diffusion models.
- A gradient-based guidance mechanism constraining generation to anatomically plausible solutions.
- Extensive validation across modalities and five segmentation architectures.
- Demonstration that pretrained natural image models can enhance medical images when properly constrained by classical priors.

## 2. Methodology

### 2.1. Framework Overview

Our hybrid enhancement framework synergistically combines classical preprocessing with pretrained diffusion models through gradient-guided sampling, eliminating the need for domain-specific training while ensuring anatomical fidelity. The framework operates through

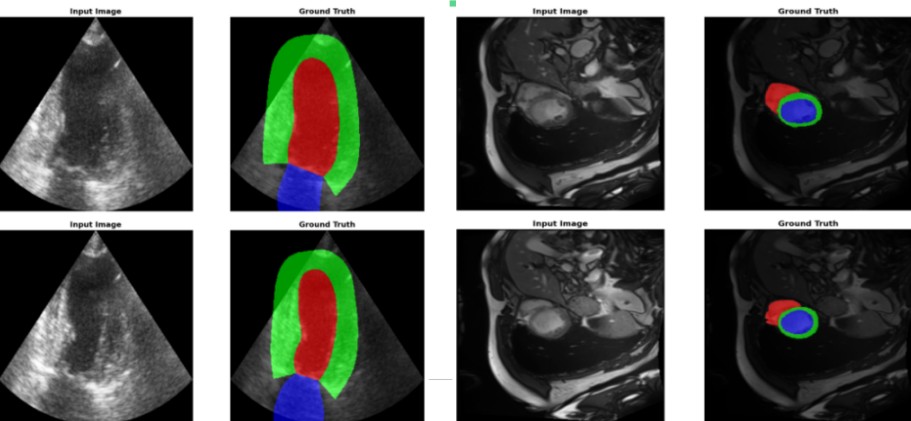

Figure 1: Representative samples from evaluation datasets. CAMUS (columns 1-2): 2D echocardiography with left ventricle (red), myocardium (green), and left atrium (blue) annotations. ACDC (columns 3-4): Cardiac MRI with right ventricle (blue), myocardium (green), and left ventricle (red) annotations.

a two-stage inference pipeline: (1) **Classical Enhancement Stage** applies modality-specific preprocessing to generate pseudo-labels that serve as guidance targets, and (2) **Diffusion Refinement Stage** leverages the frozen Stable Diffusion model to refine structural details through gradient-guided reverse diffusion, remaining anchored to the pseudo-label constraints. By not allowing unconstrained sampling, we prevent the diffusion model from hallucinating anatomically implausible structures which is a paramount concern in clinical deployment. Figure 1 shows a few samples from the CAMUS and ACDC dataset along with annotations, while Figure 2 shows the framework.

The key innovation lies in treating classical methods not as competing alternatives but as complementary guidance. Classical methods excel at removing modality-specific artifacts (speckle in ultrasound, Rician noise in MRI) and provide reliable anatomical structure, but over-smooth fine details due to hand-crafted priors. Conversely, pretrained diffusion models have learned rich natural image priors from large-scale datasets, enabling detail refinement, but lack medical domain specificity and risk generating unrealistic structures when unconstrained. Our gradient guidance mechanism bridges this gap: the diffusion model refines structures within a constrained solution space defined by classical preprocessing. Specifically, pretrained Stable Diffusion is selected because low-level features (edges, textures, gradients) learned from large-scale natural image datasets generalize across domains, and our gradient-based constraint explicitly anchors generation to the medical domain through classical preprocessing, effectively bridging the domain gap without requiring medical-specific pretraining.

## 2.2. Classical Enhancement: Modality-Specific Pseudo-Label Generation

### 2.2.1. ULTRASOUND ENHANCEMENT (SRAD → CLAHE)

Ultrasound suffers from multiplicative speckle and low contrast. We generate pseudo-labels via SRAD followed by CLAHE.

**SRAD**: Speckle-adapted anisotropic diffusion (Yu and Acton, 2002) evolves

$$\frac{\partial I}{\partial t} = \nabla \cdot [c(q)\nabla I], \quad c(q) = \frac{1}{1 + \frac{q^2 - q_0^2}{q_0^2(1+q_0^2)}},$$

(1)

with coefficient of variation $q$ and $q_0 = 50$. We run $T = 25$ explicit steps with $\Delta t = 0.02$.

**CLAHE**: Local contrast is restored via tile-wise histogram equalization with clip limit 2.0 (Zuiderveld, 1994).

$$\mathbf{y}_{\text{pseudo}}^{\text{US}} = \text{CLAHE}(\text{SRAD}(I_0)).$$

(2)

### 2.2.2. MRI ENHANCEMENT (N4ITK → NLM → CLAHE)

MRI degradation arises from bias-field inhomogeneity and Rician noise. We apply N4ITK, NLM, and CLAHE.

**N4ITK**: MRI is modeled as $I_0 = b\,u + n$. N4ITK (Tustison et al., 2010) iteratively estimates

$$I_{\text{N4}} = \frac{I_0}{\hat{b}},$$

(3)

where $\hat{b}$ is a B-spline–parameterized bias field fitted in the log domain (shrink factor 4).

**NLM**: Magnitude MRI yields Rician noise; denoising uses non-local similarity (Buades et al., 2011):

$$I_{\text{NLM}}(\mathbf{x}) = \frac{\sum_{\mathbf{y}} w(\mathbf{x}, \mathbf{y})\, I_{\text{N4}}(\mathbf{y})}{\sum_{\mathbf{y}} w(\mathbf{x}, \mathbf{y})}, \quad w = \exp\left(-\frac{\|P_x - P_y\|^2}{h^2}\right),$$

(4)

with noise-adapted bandwidth (Coupé et al., 2008)

$$h(\mathbf{x}) = \beta\sigma\sqrt{\max(1, I_{\text{N4}}(\mathbf{x})/\sigma)}, \; \beta = 1.2.$$

(5)

**CLAHE**: Final local contrast enhancement is applied to sharpen cardiac boundaries:

$$\mathbf{y}_{\text{pseudo}}^{\text{MRI}} = \text{CLAHE}(\text{NLM}(\text{N4}(I_0))).$$

(6)

Both pipelines produce anatomically consistent pseudo-labels used as soft targets for diffusion-guided refinement.

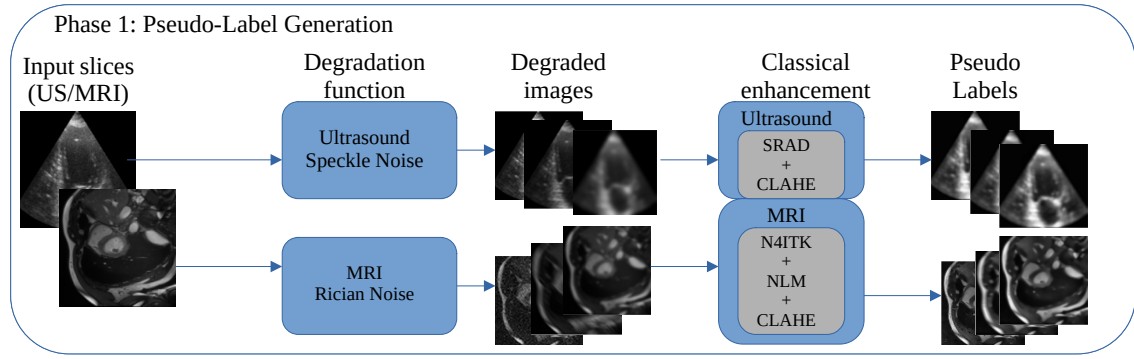

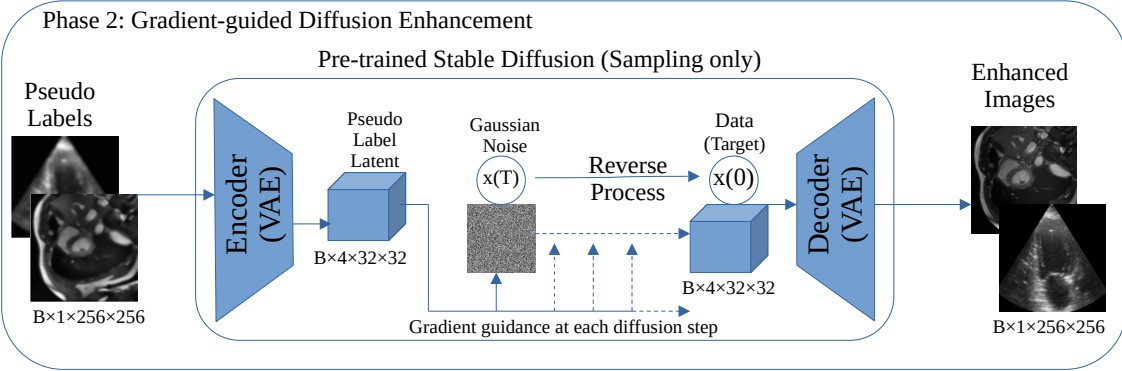

Figure 2: Two-phase hybrid enhancement framework combining classical priors with pre-trained diffusion models. Classical preprocessing generates pseudo-labels through modality-specific pipelines. These pseudo-labels provide differentiable guidance targets during reverse diffusion sampling in latent space ($256 \times 256 \rightarrow 32 \times 32 \times 4$ compression). Gradient-based guidance at each denoising step constrains the pre-trained Stable Diffusion model toward pseudo-labels, preventing hallucinations while enabling learned refinement of fine structures. The framework requires no domain-specific training, only inference-time sampling with frozen pretrained components (VAE encoder/decoder, UNet)

## 2.3. Gradient-Guided Diffusion Sampling

### 2.3.1. Latent Encoding via Stable Diffusion VAE

To enable efficient diffusion, we operate in Stable Diffusion's latent space ([Rombach et al., 2022]). A $256 \times 256$ grayscale image $\mathbf{x}$ is replicated to RGB and normalized:

$$\mathbf{x}_{\mathrm{norm}} = 2\,\mathrm{repeat}(\mathbf{x}, 3) - 1.$$

The pretrained VAE encodes

$$\mathbf{z} = 0.18215\,\mathcal{E}_{\mathrm{VAE}}(\mathbf{x}_{\mathrm{norm}}) \in \mathbb{R}^{4 \times 32 \times 32},$$

a $16\times$ spatial reduction with fixed latent scaling. Both input and pseudo-label use the same mapping:

$$\mathbf{z}_{\text{in}} = 0.18215\,\mathcal{E}_{\text{VAE}}(2\mathbf{x} - 1), \qquad \mathbf{z}_{\text{pseudo}} = 0.18215\,\mathcal{E}_{\text{VAE}}(2\mathbf{y}_{\text{pseudo}} - 1).$$

Decoding applies the inverse scale:

$$\hat{\mathbf{x}} = \tfrac{1}{2}\left(\mathcal{D}_{\text{VAE}}(\mathbf{z}/0.18215) + 1\right), \qquad \hat{\mathbf{x}}_{\text{gray}} = \tfrac{1}{3}\sum_{c=1}^{3}\hat{\mathbf{x}}_c.$$

The VAE remains frozen, its pretrained latent manifold provides a compact, stable representation for guided diffusion without domain-specific finetuning.

### 2.3.2. Gradient-Based Guidance Toward Pseudo-Labels

The key innovation of our framework is applying gradient-based guidance at each diffusion step to constrain generation toward the pseudo-label, inspired by classifier guidance (Dhariwal and Nichol, 2021) and underwater image enhancement diffusion priors (Du et al., 2025), but adapted for medical imaging.

At each reverse diffusion step $t \in \mathcal{T}$, after predicting $\hat{\mathbf{z}}_0$, we compute guidance loss $\mathcal{L}_{\text{guide}} = \|\hat{\mathbf{z}}_0 - \mathbf{z}_{\text{pseudo}}\|^2$ and apply the gradient to obtain:

$$\hat{\mathbf{z}}_0^{\text{guided}} = \hat{\mathbf{z}}_0 - \lambda\nabla_{\hat{\mathbf{z}}_0}\mathcal{L}_{\text{guide}} = (1 - 2\lambda)\hat{z}_0 + 2\lambda z_{pseudo} \tag{7}$$

where $\lambda$ is the guidance scale. The DDIM update proceeds with this guided prediction:

$$\mathbf{z}_{t-1} = \sqrt{\bar{\alpha}_{t-1}}\hat{\mathbf{z}}_0^{\text{guided}} + \sqrt{1 - \bar{\alpha}_{t-1}}\boldsymbol{\epsilon}_\theta(\mathbf{z}_t, t) \tag{8}$$

This mechanism acts as a **soft constraint** that pulls the diffusion model toward the pseudo-label while allowing refinement through learned priors. We use $\lambda = 1200$ for ultrasound and $\lambda = 1000$ for MRI, determined empirically. This soft constraint design also provides robustness to pseudo-label quality: $\lambda$ controls the trust placed in pseudo-labels, so if classical preprocessing produces imperfect results on severely degraded inputs, the diffusion model can rely more on its learned priors by reducing $\lambda$, preventing catastrophic guidance toward incorrect anatomical targets. Algorithm 1 shows the steps of the framework.

### 2.4. Realistic Degradation Simulation

For Ultrasound, we apply synthetic degradation to each image as **Speckle noise** which is a multiplicative noise modeling coherent interference,
$I_{\text{speckle}} = \text{clip}(I_0 \cdot (1 + \eta), 0, 1)$ where $\eta \sim \mathcal{N}(0, \sigma_n^2)$ with $\sigma_n \in [0.1, 0.3]$ sampled uniformly.
For MRI, we apply synthetic degradation to each image as **Rician noise** which is the magnitude reconstruction noise from complex MRI signals,
$I_{\text{Rician}} = \sqrt{(I_0 + \eta_{\text{real}})^2 + \eta_{\text{imag}}^2}$ where $\eta_{\text{real}}, \eta_{\text{imag}} \sim \mathcal{N}(0, \sigma_n^2)$ with $\sigma_n \in [0.02, 0.04]$ sampled uniformly.

---

**Algorithm 1:** Gradient-Guided Diffusion Enhancement

---

**Input:** Image $\mathbf{x}$, guidance scale $\lambda$, timesteps $\mathcal{T}$

**Output:** Enhanced image $\hat{\mathbf{x}}$

$\mathbf{y}_{\text{pseudo}} \leftarrow \text{ClassicalEnhance}(\mathbf{x})$;

$\mathbf{z}_{\text{pseudo}} \leftarrow 0.18215 \, \mathcal{E}_{VAE}(2\mathbf{y}_{\text{pseudo}} - 1)$;

Sample $\mathbf{z}_T \sim \mathcal{N}(0, I)$;

**for** $t$ *in* $\mathcal{T}$ *(reverse)* **do**

    $\epsilon_t \leftarrow \epsilon_\theta(\mathbf{z}_t, t)$;

    $\hat{\mathbf{z}}_0 \leftarrow (\mathbf{z}_t - \sqrt{1 - \bar{\alpha}_t} \, \epsilon_t)/\sqrt{\bar{\alpha}_t}$;

    $\hat{\mathbf{z}}_0 \leftarrow \hat{\mathbf{z}}_0 - 2\lambda(\hat{\mathbf{z}}_0 - \mathbf{z}_{\text{pseudo}})$;

    **if** $t > 0$ **then**

        $\mathbf{z}_{t-1} \leftarrow \sqrt{\bar{\alpha}_{t-1}}\hat{\mathbf{z}}_0 + \sqrt{1 - \bar{\alpha}_{t-1}}\epsilon_t$;

    **end**

**end**

$\hat{\mathbf{x}} \leftarrow \frac{1}{2}(\mathcal{D}_{VAE}(\mathbf{z}_0/0.18215) + 1)$;

$\hat{\mathbf{x}} \leftarrow \text{RGB2Gray}(\hat{\mathbf{x}})$;

**return** $\hat{\mathbf{x}}$;

---

### 2.5. Downstream Segmentation Evaluation

While image quality metrics (PSNR, SSIM) provide quantification, the ultimate test is whether enhancement improves clinical task performance. For cardiac imaging, accurate segmentation of left ventricle, myocardium, and atrium is critical for diagnosis. We evaluate our method by measuring downstream segmentation accuracy across five popular architectures: **U-Net** (Ronneberger et al., 2015), **Attention U-Net** (Oktay et al., 2018), **UNETR** (Hatamizadeh et al., 2022), **DeepLabV3+** (Chen et al., 2018), and **U-Net++** (Zhou et al., 2018). This diversity ensures our findings generalize across different architectural paradigms (CNNs, attention mechanisms, transformers).

### 3. Training Protocol

For each architecture, we train the model on clean images to get a pretrained segmentation model. Then we test on three variants of the test data: **Baseline** (original images degraded through modality specific degradation $I_{\text{deg}}$), **Classical** (pseudo-labels $\mathbf{y}_{\text{pseudo}}$), and **Ours** (diffusion-enhanced $\hat{\mathbf{x}}_{\text{enhanced}}$). All models use identical hyperparameters: Adam optimizer ($\beta_1 = 0.9$, $\beta_2 = 0.999$), learning rate $10^{-4}$ with cosine annealing, batch size 16, 500 epochs with early stopping (patience 20), combined Dice and cross-entropy loss $\mathcal{L} = \mathcal{L}_{\text{Dice}} + \mathcal{L}_{\text{CE}}$, and standard augmentations (rotation $\pm 15$, scaling 0.9–1.1, flipping). This ensures fair comparison across image variants. We additionally evaluate a realistic scenario where segmentation models are trained on degraded images and tested on our enhanced outputs, to simulate clinical conditions where clean training data is unavailable.

## 3.1. Evaluation Metrics

We evaluate segmentation using Dice, IoU, Average Surface Distance (ASD), and the 95th percentile Hausdorff Distance (HD95). For quantifying enhancement we use the NIQE metric. Let $P_c$ and $G_c$ denote predicted and ground-truth masks for class $c$.

**Dice:**

$$\text{Dice}_c = \frac{2|P_c \cap G_c|}{|P_c| + |G_c|}. \tag{9}$$

**IoU:**

$$\text{IoU}_c = \frac{|P_c \cap G_c|}{|P_c \cup G_c|}. \tag{10}$$

**ASD:**

$$\text{ASD}(P_c, G_c) = \frac{1}{|S_P| + |S_G|} \left( \sum_{p \in S_P} d(p, S_G) + \sum_{g \in S_G} d(g, S_P) \right), \tag{11}$$

where $S_P, S_G$ are surface voxels.

**HD95:**

$$\text{HD95}(P_c, G_c) = \text{percentile}_{95}(\{d(p, S_G)\}_{p \in S_P} \cup \{d(g, S_P)\}_{g \in S_G}). \tag{12}$$

**NIQE:**

$$\text{NIQE}(I) = \sqrt{(\mathbf{v} - \mathbf{v}_{\text{NSS}})^T \left( \frac{\Sigma_1 + \Sigma_2}{2} \right)^{-1} (\mathbf{v} - \mathbf{v}_{\text{NSS}})}, \tag{13}$$

where $\mathbf{v}$ are natural scene statistics features extracted from image $I$, $\mathbf{v}_{\text{NSS}}$ are features from the natural image database, and $\Sigma_1, \Sigma_2$ are their respective covariance matrices. Lower NIQE indicates better perceptual quality.

## 3.2. Experimental Design

### 3.2.1. DATASETS

**CAMUS (Ultrasound):** The Cardiac Acquisitions for Multi-structure Ultrasound Segmentation dataset (Leclerc et al., 2019) contains 2D echocardiography from 500 patients with end-diastolic and end-systolic frames ($256 \times 256$ pixels). Expert annotations include left ventricle, myocardium, and left atrium. We use 450 patients for training, 50 for testing as provided in the dataset.

**ACDC (MRI):** The Automated Cardiac Diagnosis Challenge dataset (Bernard et al., 2018) contains short-axis cardiac cine-MRI from 150 patients across five pathology groups (Dilated cardiomyopathy, Hypertrophic cardiomyopathy, Myocardial infarction , Abnormal right ventricle ) plus healthy controls. Annotations include right ventricle, myocardium, and left ventricle ($256 \times 256$ pixels). We use 100 patients for training, 50 for testing as provided in the dataset.

We implement our framework in PyTorch 2.13 with HuggingFace Diffusers, MONAI, and OpenCV libraries. All experiments run on NVIDIA H200 GPUs (141GB). Inference time for reverse diffusion (10 steps) is approximately 5-10 seconds per image and is largely

dependent upon pseudo label generation phase, which is acceptable for non-emergency clinical workflows.

**Reproducibility:** Code is available at https://github.com/pks716/MIDL_26.

## 4. Results

### 4.1. Segmentation Performance

Tables 1 and 2 present comprehensive segmentation results. Our method consistently outperforms both degraded baselines and classical preprocessing across all architectures and modalities.

Table 1: Segmentation performance comparison across architectures and enhancement methods on CAMUS ultrasound dataset. Metrics reported are averaged across cardiac structures (LV cavity, myocardium, LA cavity), excluding background. Bold indicates best performance per architecture. Variation in NIQE score is due to degradation sampling being random.

| Architecture | Dice ↑ | | | IoU ↑ | | | HD95 (px) ↓ | | | ASD (px) ↓ | | | NIQE ↓ | | |
|---|---|---|---|---|---|---|---|---|---|---|---|---|---|---|---|
| | Degr. | Class. | **Ours** | Degr. | Class. | **Ours** | Degr. | Class. | **Ours** | Degr. | Class. | **Ours** | Degr. | Class. | **Ours** |
| U-Net | 0.822 | 0.839 | **0.859** | 0.712 | 0.727 | **0.746** | 21.41 | 20.20 | **18.89** | 9.15 | 7.35 | **6.55** | 11.19 | 5.91 | **5.33** |
| | ±0.089 | ±0.067 | **±0.054** | ±0.104 | ±0.081 | **±0.067** | ±2.31 | ±1.87 | **±1.52** | ±0.67 | ±0.52 | **±0.41** | ±1.24 | ±1.03 | **±0.97** |
| Attention U-Net | 0.831 | 0.844 | **0.862** | 0.710 | 0.723 | **0.741** | 20.33 | 20.19 | **19.18** | 8.81 | 7.22 | **5.31** | 11.22 | 5.63 | **5.35** |
| | ±0.086 | ±0.065 | **±0.053** | ±0.102 | ±0.079 | **±0.066** | ±2.25 | ±1.83 | **±1.49** | ±0.66 | ±0.51 | **±0.40** | ±1.22 | ±1.01 | **±0.96** |
| UNETR | 0.830 | 0.839 | **0.854** | 0.721 | 0.725 | **0.744** | 20.35 | 19.58 | **19.10** | 8.55 | 7.35 | **5.65** | 10.35 | 5.75 | **5.61** |
| | ±0.087 | ±0.066 | **±0.054** | ±0.103 | ±0.080 | **±0.067** | ±2.27 | ±1.84 | **±1.50** | ±0.66 | ±0.51 | **±0.40** | ±1.21 | ±1.01 | **±0.96** |
| DeepLabV3+ | 0.845 | 0.859 | **0.868** | 0.747 | 0.758 | **0.771** | 19.76 | 19.10 | **18.49** | 8.12 | 7.18 | **6.22** | 10.53 | 6.18 | **5.45** |
| | ±0.083 | ±0.062 | **±0.050** | ±0.099 | ±0.076 | **±0.063** | ±2.16 | ±1.73 | **±1.38** | ±0.63 | ±0.49 | **±0.38** | ±1.18 | ±0.98 | **±0.93** |
| U-Net++ | 0.841 | 0.852 | **0.863** | 0.745 | 0.753 | **0.772** | 19.98 | 18.87 | **18.41** | 8.24 | 7.71 | **6.37** | 10.69 | 6.35 | **5.80** |
| | ±0.084 | ±0.063 | **±0.051** | ±0.100 | ±0.077 | **±0.064** | ±2.19 | ±1.76 | **±1.41** | ±0.64 | ±0.50 | **±0.39** | ±1.20 | ±0.99 | **±0.94** |

Table 2: Segmentation performance comparison across architectures and enhancement methods on ACDC cardiac MRI dataset. Metrics reported are averaged across cardiac structures (RV cavity, myocardium, LV cavity), excluding background. Bold indicates best performance per architecture. Variation in NIQE score is due to degradation sampling being random.

| Architecture | Dice ↑ | | | IoU ↑ | | | HD95 (px) ↓ | | | ASD (px) ↓ | | | NIQE ↓ | | |
|---|---|---|---|---|---|---|---|---|---|---|---|---|---|---|---|
| | Degr. | Class. | **Ours** | Degr. | Class. | **Ours** | Degr. | Class. | **Ours** | Degr. | Class. | **Ours** | Degr. | Class. | **Ours** |
| U-Net | 0.729 | 0.752 | **0.785** | 0.637 | 0.672 | **0.701** | 11.57 | 10.87 | **9.42** | 5.67 | 5.03 | **4.61** | 8.45 | 7.32 | **7.11** |
| | ±0.095 | ±0.071 | **±0.058** | ±0.112 | ±0.086 | **±0.072** | ±2.56 | ±1.98 | **±1.67** | ±0.73 | ±0.58 | **±0.47** | ±1.38 | ±1.15 | **±1.06** |
| Attention U-Net | 0.802 | 0.829 | **0.847** | 0.723 | 0.741 | **0.760** | 11.16 | 10.45 | **9.28** | 5.11 | 4.89 | **4.50** | 8.23 | 7.89 | **6.74** |
| | ±0.088 | ±0.066 | **±0.053** | ±0.104 | ±0.080 | **±0.066** | ±2.38 | ±1.84 | **±1.54** | ±0.68 | ±0.54 | **±0.44** | ±1.31 | ±1.09 | **±1.01** |
| UNETR | 0.741 | 0.754 | **0.768** | 0.649 | 0.698 | **0.717** | 10.31 | 10.02 | **9.83** | 5.89 | 5.21 | **4.82** | 8.78 | 7.45 | **6.92** |
| | ±0.101 | ±0.078 | **±0.065** | ±0.118 | ±0.093 | **±0.079** | ±2.78 | ±2.15 | **±1.89** | ±0.79 | ±0.63 | **±0.54** | ±1.45 | ±1.23 | **±1.15** |
| DeepLabV3+ | 0.812 | 0.828 | **0.837** | 0.731 | 0.739 | **0.748** | 9.48 | 9.12 | **8.81** | 5.60 | 4.98 | **4.57** | 8.71 | 8.05 | **7.51** |
| | ±0.092 | ±0.069 | **±0.056** | ±0.108 | ±0.084 | **±0.070** | ±2.48 | ±1.92 | **±1.63** | ±0.71 | ±0.56 | **±0.46** | ±1.35 | ±1.12 | **±1.04** |
| U-Net++ | 0.824 | 0.831 | **0.836** | 0.742 | 0.751 | **0.767** | 8.89 | 8.58 | **8.18** | 5.54 | 4.93 | **4.47** | 8.31 | 6.98 | **6.45** |
| | ±0.090 | ±0.068 | **±0.055** | ±0.106 | ±0.082 | **±0.069** | ±2.43 | ±1.87 | **±1.59** | ±0.69 | ±0.55 | **±0.45** | ±1.33 | ±1.11 | **±1.03** |

## 4.2. Qualitative Results

Figure 3 shows representative segmentation examples. For ultrasound, our method removes speckle while preserving myocardial texture and recovers fine structures. For MRI, classical method corrects bias fields but struggles with severe Rician noise in low-SNR regions; our method refines these areas while preserving tissue intensity relationships.

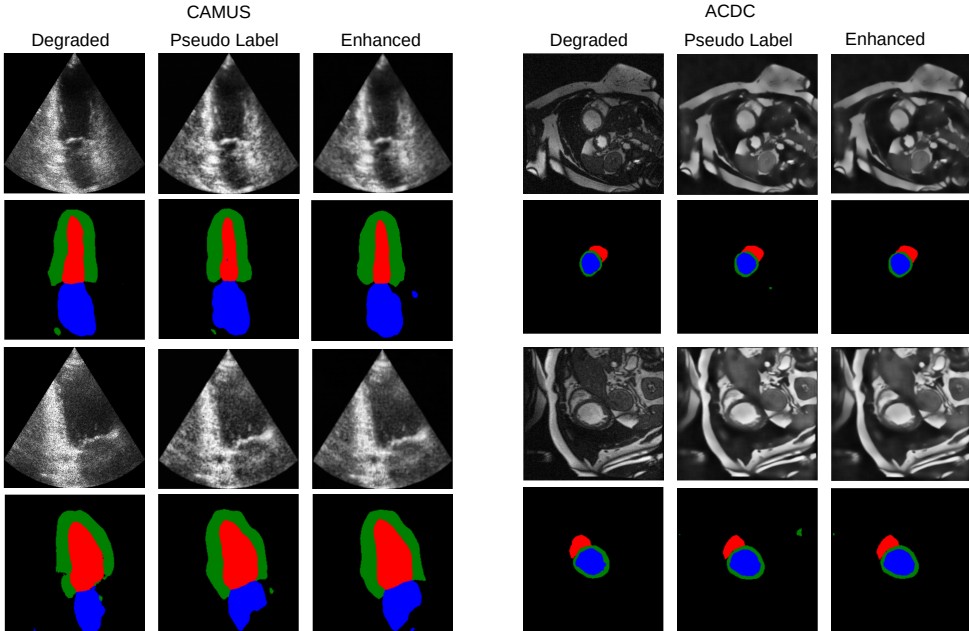

Figure 3: Representative samples from test datasets. CAMUS (columns 1-3): Cardiac ultrasound showing degraded, pseudo label, enhanced images and their respective segmentation results. ACDC (columns 4-6): Cardiac MRI showing degraded, pseudo label, enhanced images and their respective segmentation results.

## 4.3. Ablation Study

Table 3 presents an ablation on CAMUS (U-Net) isolating the contribution of pseudo-label guidance. Unconstrained diffusion collapses without guidance (Dice 0.484, hallucinations), confirming that pseudo-labels are essential as differentiable guidance targets.

## 4.4. Supervised Baseline Comparison

We trained a supervised Diffusion model on CAMUS (450 patients, 100k steps, ∼12 GPU-hours on H200, AdamW lr=$10^{-5}$ with cosine decay, L2 reconstruction loss) as a direct comparison. Our training-free method achieves Dice $0.863 \pm 0.051$ vs. supervised diffusion $0.861 \pm 0.072$, matching supervised performance without any training cost or overfitting risk on limited medical data.

Table 3: Ablation study on CAMUS dataset showing the necessity of pseudo-label guidance.

| Method (U-Net) | Dice |
|---|---|
| Classical pipeline alone | 0.839 |
| Diffusion without pseudo-label guidance | 0.484 |
| Ours (diffusion + pseudo-label guidance) | **0.859** |

### 4.5. Realistic Training Scenario

Table 4 reports results when the segmentation model is trained on degraded images (CA-MUS, U-Net), simulating real-world clinical deployment where clean training data is unavailable. Enhancement benefits persist in this scenario, confirming genuine structural recovery.

Table 4: Realistic training scenario: segmentation model trained on degraded images.

| Training Data | Test Data | Dice |
|---|---|---|
| Degraded | Degraded | 0.841 |
| Degraded | Ours (enhanced) | **0.862** |

### 4.6. Intensity Distribution Analysis

Table 5 confirms our method introduces minimal intensity shift relative to classical preprocessing on the CAMUS test set (values normalized to $[0, 1]$), while improving boundary accuracy.

Table 5: Intensity distribution analysis on CAMUS test set.

| Method | Mean Intensity | Std (Variability) |
|---|---|---|
| Degraded | $0.190 \pm 0.036$ | $0.231 \pm 0.034$ |
| Classical (Pseudo-label) | $0.227 \pm 0.024$ | $0.247 \pm 0.020$ |
| Ours | $0.219 \pm 0.024$ | $0.239 \pm 0.020$ |

## 5. Discussion

### 5.1. Key Findings

Our hybrid framework demonstrates that pretrained diffusion models can enhance medical images when properly constrained by classical domain priors. Three key findings emerge: (1) gradient-guided sampling prevents hallucinations while enabling learned refinement; (2) the framework generalizes across modalities, anatomical structures, and deep learning architectures, without requiring domain-specific training; (3) boundary accuracy improvements

(HD95 and ASD reduction) indicate enhanced fine structure preservation critical for clinical measurements.

## 5.2. Limitations and Future Work

The optimal guidance scale $\lambda$ is modality-dependent and determined empirically; future work should explore adaptive guidance schedules that adjust $\lambda$ per-image based on degradation severity or anatomical region, potentially improving robustness across diverse clinical scenarios. Current work processes 2D slices independently, which is intentional: CAMUS provides only 2D frames, and ACDC is evaluated slice-wise in the official challenge. This unified 2D design reduces memory requirements (4GB vs. 16–32GB for 3D diffusion models) and generalizes to inherently 2D modalities (X-ray, single-slice CT), though extending to 3D could leverage inter-slice consistency at the cost of substantially higher memory requirements. The current pipeline runs in 5–10 seconds per image (3–6s classical preprocessing, 2–4s diffusion sampling), targeting offline post-acquisition workflows such as ejection fraction analysis and clinical reporting rather than real-time scanning; distillation-based acceleration could enable faster deployment in latency-sensitive settings. Evaluation on synthetically degraded images using physics-derived noise models (multiplicative speckle for ultrasound, Rician for MRI) follows established practice due to the clinical infeasibility of acquiring paired clean/degraded data, and real-world validation with radiologist evaluation remains an important avenue for future work. Validation on additional modalities (CT, X-ray, microscopy) would further establish generalizability, with each modality potentially requiring custom classical pipelines while the core gradient guidance mechanism transfers.

## 5.3. Conclusion

We presented a training-free hybrid framework that bridges classical preprocessing with pretrained diffusion models for medical image enhancement. Gradient-based guidance toward modality-specific pseudo-labels ensures anatomical fidelity while enabling learned refinement, achieving consistent improvements across ultrasound and MRI datasets without domain-specific training. This work establishes that pretrained natural image models, when properly constrained by classical domain priors, can enhance medical images while preserving the interpretability essential for clinical adoption.

## Acknowledgments

This work was supported by ICMR (Grant ID: FIW-2024-01-00000151),
Project No: IITM/ICMR/SS/537, IIT Mandi.
The authors gratefully acknowledge Dr. Aditya Nigam (IIT Mandi) for his supervision, research guidance, and provision of computational infrastructure. The authors also acknowledge Dr. Pankaj Gupta (PGIMER, Chandigarh) for his expert contributions on medical imaging modalities, clinical workflow considerations, and diagnostic requirements that informed the clinical relevance of this work.

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
