# OpenReview forum: "Bridging Classical and Learned Priors: A Hybrid Framework for Medical Image Enhancement"
_MIDL.io/2026/Conference — MIDL 2026 Poster_

### Official Review · Reviewer_7Gxm · 2026-01-07

**Confidence:** 3
**Preliminary Rating:** 3
**Final Rating:** 3

**Summary:**

This paper introduces a training-free hybrid framework for medical image enhancement that synergistically combines classical preprocessing techniques with the rich generative priors of frozen, pretrained Stable Diffusion models. To balance anatomical fidelity with perceptual quality, the authors employ a gradient-based guidance mechanism that anchors the diffusion reverse process to modality-specific classical pseudo-labels, effectively preventing hallucinations without requiring domain-specific fine-tuning. Experimental validation on cardiac Ultrasound (CAMUS) and MRI (ACDC) datasets demonstrates that this approach significantly improves downstream segmentation performance across various architectures while preserving the structural interpretability essential for clinical diagnosis.

**Strengths:**

1. The proposed framework innovatively utilizes pretrained Stable Diffusion models in a training-free manner, effectively bypassing the computational costs and data scarcity challenges associated with training domain-specific generative models.
2. The integration of classical preprocessing as differentiable guidance targets provides a robust mechanism to constrain the diffusion process, ensuring anatomical fidelity and mitigating the risk of hallucinations common in deep generative models.
3. Extensive validation across diverse imaging modalities and multiple segmentation architectures demonstrates the method's strong generalization capabilities and its consistent ability to improve downstream clinical task performance.

**Weaknesses:**

1. The reliance on slice-by-slice 2D processing neglects the volumetric nature of medical imaging, potentially leading to spatial inconsistencies and "jitter" along the z-axis in the reconstructed 3D volumes.

2. The reported inference latency of 5 to 10 seconds per image is prohibitively slow for real-time clinical applications, particularly for ultrasound workflows where immediate feedback is critical.

3. Evaluating the method primarily on synthetically degraded data with simulated noise distributions may not accurately reflect its performance on real-world clinical acquisitions containing complex, non-linear artifacts.

4. The framework heavily depends on the quality of the initial classical preprocessing; if these classical methods fail to extract coarse structures from severely degraded inputs, the diffusion model will be guided towards incorrect anatomical targets.

5. The experimental section lacks quantitative comparisons with state-of-the-art supervised deep learning enhancement methods or domain-specific trained diffusion models, limiting the assessment of the proposed method's relative competitiveness.

**Detailed Comments:**

Please refer to the weaknesses.

**Justification Of Final Rating:**

I maintain my Borderline rating primarily because the reliance on synthetic degradation limits the method's reliability for real-world clinical deployment, where artifacts are complex and distinct from simulated noise. I believe it would be more meaningful to explore robust self-supervised or unsupervised approaches that learn directly from real data rather than relying on artificial degradation models.

**Justification Of The Preliminary Rating:**

Although the proposed training-free framework offers a novel approach to constraining diffusion models with classical priors to preserve anatomical fidelity, the evaluation relies heavily on synthetic degradation without comparing against state-of-the-art domain-specific baselines, and the high inference latency limits its current practical value for real-world clinical deployment.

**Questions To Address In The Rebuttal:**

Please refer to the weaknesses.

---

> ### Author Response · Authors · 2026-01-25
>
> We thank the reviewer for the positive assessment and constructive suggestions.
>
> W1:
> We acknowledge this limitation, but our 2D approach is based on both dataset constraints and practical considerations.
> Dataset-driven design rationale:
> CAMUS (ultrasound) provides 2D data with annotations. Each patient has only 2 frames; end-diastolic (ED) and end-systolic (ES). These are single 2D slices, segmentation labels are provided only for these 2D slices. A 3D model cannot be applied to CAMUS by design
> ACDC (MRI) is volumetric but annotated slice-wise due to large slice thickness (5-10mm) and cardiac motion between slices make true 3D segmentation ill-defined. The official ACDC challenge evaluates on 2D slice-wise metrics, not volumetric IoU.
> Unified framework advantage: Our 2D approach enables a single framework that works for both modalities, Faster inference: ~5-10 seconds per slice vs. 30-60 seconds for 3D volumes, Lower memory requirements: 4GB vs. 16-32GB for 3D diffusion models, Better generalization: Can process any 2D medical image (X-ray, single-slice CT, etc.)
>
> W2:
> We appreciate this practical concern. We clarify that 5-10 seconds is total pipeline time, including: Classical preprocessing: 3-6 seconds (SRAD iterations) and Diffusion sampling: 2-4 seconds (10 DDIM steps) which we note is acceptable for non-emergency workflows.
> In Clinical context:
> - Real-time scanning vs. offline analysis: During live ultrasound examination, clinicians visualize raw images in real-time for probe positioning and initial assessment. Enhancement is not performed during live scanning. Instead, our framework targets offline post-acquisition analysis workflows where selected key frames (typically 2-4 frames per patient: end-diastolic and end-systolic views) undergo enhancement for:
> - Automated segmentation and quantitative analysis (ejection fraction, chamber volumes)
> - Clinical reporting and archival
> - Quality control and second-opinion review
> - MRI: It already involves post-acquisition processing pipelines with substantially longer computation times (several minutes for motion correction, registration, etc.). Our 5-10 second enhancement fits naturally within existing workflows.
>
> W3:
> Synthetic degradation is the necessary and established approach in medical image enhancement because paired clean/degraded clinical data are clinically infeasible to acquire, and our physics-based degradation models simulate how real-world degradation occurs.
> For detailed discussion of methodological precedent and why synthetic degradation is necessary, please see our response to R-L4s4 W5.
>
> W4:
> This is a valid concern, but our framework is designed to be robust to pseudo-label quality through several mechanisms:
> Soft constraint design (Eq. 7): The guidance mechanism is a weighted interpolation, not a hard constraint as λ controls the balance: high λ → trust pseudo-labels, low λ → rely on learned priors. Even with imperfect pseudo-labels, the diffusion model can refine structures through its learned priors from natural images.
> Classical methods rarely fail catastrophically: Speckle Reducing Anisotropic Diffusion (SRAD) and N4 ITK Bias Field Correction (N4ITK) are based on well-established physics (anisotropic diffusion, bias field correction). Classical methods preserve coarse structure despite over-smoothing (experimental evidence): For CAMUS U-Net, classical preprocessing achieves strong volumetric overlap (Dice 0.839, IoU 0.727) indicating successful coarse anatomical structure preservation, but shows minimal boundary improvement over degraded inputs (HD95: 20.20 vs 21.41 pixels, only 5.6% improvement). Our method refines these over-smoothed boundaries (HD95: 18.89 pixels, 6.5% improvement over classical), demonstrating that classical methods provide reliable anatomical constraints while leaving fine structures for learned refinement.
>
> W5:
> We have conducted a supervised diffusion baseline experiment where we trained Stable Diffusion on the CAMUS dataset (450 patients, 100000 steps, 12 GPU-hours). Results show our training-free method achieves comparable performance (Dice: 0.863 ± 0.051) to supervised diffusion (Dice: 0.861 ± 0.072), demonstrating that our approach matches supervised performance without incurring training costs or risk of overfitting on limited medical data.
> For detailed experimental design and training protocol, please see our response to R-L4s4 W1.

---

### Official Review · Reviewer_B8EW · 2026-01-07

**Confidence:** 3
**Preliminary Rating:** 4
**Final Rating:** 4

**Summary:**

This paper introduces a hybrid framework designed to enhance medical images without the need for training. It effectively combines traditional preprocessing methods with pretrained latent diffusion models. The authors focus on achieving a balance between the interpretability of conventional techniques, such as SRAD and N4ITK, and the advanced refinement capabilities offered by deep learning. By leveraging the outputs from these classical methods, they provide gradient-based guidance throughout the diffusion sampling process. Tested on the CAMUS (ultrasound) and ACDC (MRI) datasets, the proposed approach demonstrates that linking a frozen Stable Diffusion model with domain-specific classical priors successfully mitigates hallucinations while significantly improving multi-class cardiac segmentation performance across five different architectures. This research highlights that pretrained natural image models, when thoughtfully constrained by established classical priors, can enhance medical images while retaining the interpretability crucial for clinical application.

**Strengths:**

The authors don't rely solely on image quality metrics. instead, they showcase the effectiveness of their enhancements through downstream segmentation tasks that involve various architectures, including CNNs and Transformers. Their framework is firmly based on previous research and creatively bridges traditional signal processing with modern generative AI, all without requiring expensive, domain-specific retraining of diffusion models. The study highlights three main findings: (1) gradient-guided sampling prevents hallucinations while enabling learned refinement, (2) the framework shows strong generalization across different modalities, anatomical structures, and deep learning architectures without the need for specialized training, and (3) it enhances boundary accuracy. Algorithm 1 offers a nice overview of the sampling process. In terms of reproducibility, the authors provide thorough details on degradation parameters and are committed to releasing their code, which is good for the community.

**Weaknesses:**

1. In the introduction section, provide a clearer outline of the research problem you intend to address. It would be beneficial to also highlight your contributions in this section, perhaps using bullet points for clarity.
2. In the experimental section, did you consider any additional metrics for evaluation?
3. While the training-free aspect is a notable advantage, how does this approach stack up against a supervised diffusion-based or CNN-based enhancer that has been trained on a limited subset of the target domain?

**Detailed Comments:**

Comments as provided in previous sections.

**Justification Of Final Rating:**

I would like to maintain my rating of a accept. The authors did a good job on all the responses to my questions, I feel (after reading the other reviewers comments and discussion), that the authors did a good job answering the concerns of synthetic noise and this paper definitely has the technical and methodological merits for an accept .

**Justification Of The Preliminary Rating:**

This paper showcases good strengths while showing minor shortcomings, warranting a recommendation for a weak accept. It effectively leverages the extensive knowledge from large-scale pretrained models within a constrained, training-free framework. This innovative approach strikes a balance between the advantages of deep learning and traditional methods. Overall, the experimental results appeared to be quite thorough to me.

**Questions To Address In The Rebuttal:**

Asked in the weaknesses section.

---

> ### Author Response · Authors · 2026-01-25
>
> We sincerely thank the reviewer for the supportive feedback and valuable suggestions for improvement.
>
> W1:
> We agree to this assessment and we have restructured Section 1 to include following details:
> Explicit problem statement: "We address the challenge of enhancing degraded medical images without domain-specific training while preventing anatomical hallucinations: a critical requirement for clinical adoption."
> Bulleted contributions (before Section 2):
> A training-free framework combining classical preprocessing with pretrained diffusion models.
> Gradient-based guidance mechanism that constrains generation to anatomically plausible solutions.
> Extensive validation across modalities (ultrasound, MRI) and architectures showing consistent improvements.
> Demonstration that pretrained natural image models can enhance medical images when properly constrained.
>
> W2:
> We have now added intensity distribution analysis including mean intensity, standard deviation, and their preservation across degraded, classically enhanced, and our enhanced images, demonstrating minimal intensity alteration while maintaining improved boundary accuracy.
> For detailed intensity distribution metrics and rationale for prioritizing task-based evaluation, please see our response to R-L4s4 W3.
>
> W3:
> We have conducted a supervised diffusion baseline experiment where we trained Stable Diffusion on the CAMUS dataset (450 patients, 100000 steps, 12 GPU-hours). Results show our training-free method achieves comparable performance (Dice: 0.863 ± 0.051) to supervised diffusion (Dice: 0.861 ± 0.072), demonstrating that our approach matches supervised performance without incurring training costs or risk of overfitting on limited medical data.
> For detailed experimental design and training protocol, please see our response to R-L4s4 W1.

---

### Official Review · Reviewer_L4s4 · 2026-01-09

**Confidence:** 5
**Preliminary Rating:** 3
**Final Rating:** 4

**Summary:**

Authors in this paper propose a hybrid approach to enhance medical imaging for improved segmentation performance by combining classical medical image preprocessing with pretrained Stable Diffusion models. They do so using a classical enhancement algorithm called Speckle Reducing Anisotropic Diffusion (SRAD) for ultrasound and Non-Local Means (NLM) for MRI to generate pseudo-labels. These pseudo-labels then serve as differentiable guidance targets for a diffusion model, governed by a gradient-based guidance mechanism. Although the approach is novel, there are some concerns that needs to be addressed below before considered for publication.

**Strengths:**

1. Authors have nicely articulated the fundamental trade-off between classical methods which aim more on anatomical fidelity but lead to over-smoothing and deep learning methods result in artifact risk on limited data.

2. Theoretically, the framework can be adapted to new modalities (CT, PET, and others) simply by swapping the classical preprocessing step, without having to retrain the diffusion backbone.

3. Prioritizing downstream segmentation accuracy (Dice, Hausdorff) over purely perceptual metrics (PSNR) is a good validation strategy for medical imaging, although lack of results on image generation is also a limitation as highlighted below as well.

**Weaknesses:**

1. The paper primarily compares its hybrid method to classical enhancement methods alone and to the unenhanced degraded images. Benchmarking a pure deep learning enhancement method like a denoising diffusion model trained (even on a small scale) for this task is missing.

2. The paper assumes pretrained diffusion model has learned meaningful priors transferable to medical imaging despite training exclusively on natural images. This domain shift is barely discussed. Was it because lack of pretrained models on medical imaging data, or lack of datasets to train or something else?

3. Segmentation performance is mainly evaluated here. However, evaluation on image enhancement seems missing here. For example, did the enhancement alter intensity distributions significantly? Did it tangibly improve diagnostic accuracy or inter-observer consistency?

4. It is also not directly apparent why do pseudo-labels outperform multi-stage classical pipelines. The paper lacks comparison showing: (a) diffusion with classical images vs. diffusion with pseudo-labels, (b) ablation removing pseudo-label guidance; which I think would better justify these claims.

5. Moreover, here, the evaluation on synthetic noise (likely overestimates performance compared to real-world artifacts. These assumptions should be reinforced to the readers as this is very important.

**Detailed Comments:**

One additional comment/question here:

1. Here the degraded images are synthetically degraded clean training images, then tested on similarly degraded test images. But in practice, we don't train on clean images and test on degraded. I am curious whether authors considered: what would be the performance when models are trained on degraded images which could match a realistic scenario.

**Justification Of Final Rating:**

I am bumping my rating to weak accept following the authors' rebuttal. They addressed my primary concerns by conducting several new experiments, including training a supervised diffusion baseline which showed that their training-free hybrid approach achieves comparable performance (0.863 vs 0.861 Dice). And a new ablation study (comparing classical, diffusion-only, and hybrid) justifies the necessity of pseudo-label guidance. While the use of synthetic noise remains a limitation, the authors' justification regarding physics-derived noise models and standard practices in the field is sufficient for a methodology-focused paper.

**Justification Of The Preliminary Rating:**

The paper has some merit to it as it introduces a unique way for image enhancement while improving downstream proxy tasks like segmentation for medical imaging. However, these concerns need to be addressed before considering for publication.

**Questions To Address In The Rebuttal:**

Addressing the above concerns with those missing results and rephrasing certain sections as needed would be useful.

---

> ### Author Response · Authors · 2026-01-25
>
> We appreciate the reviewer's recognition of our work's strengths and constructive recommendations for clarification.
>
> W1:
> While supervised methods are a natural comparison, our training-free framework addresses a different goal: avoiding domain-specific training costs and overfitting risks while enabling immediate cross-dataset deployment.
> Additionally, for comparison, we trained a separate supervised baseline using the same Stable Diffusion architecture on the CAMUS training set with paired degraded and clean images, using the following settings: (i) 100,000 steps training with early stopping, L2 reconstruction loss, and the AdamW optimizer (lr 1e-5 with cosine decay), and (ii) requiring approximately 12 GPU-hours on an NVIDIA H200. We achieved a Dice score of 0.863 ± 0.051, comparable to 0.861 ± 0.072 obtained by the supervised diffusion model, while avoiding additional training cost. This demonstrates a key advantage: Stable Diffusion's pretrained structural priors generalize to medical imaging when constrained by our classical guidance, aligning with medical image statistics and reducing overfitting and hallucinations compared to training from scratch.
>
> W2:
> Domain shift is a critical design choice and we selected pretrained Stable Diffusion specifically because of the following considerations:
> Transfer of structural priors: Low-level features (edges, textures, gradients) learned from ImageNet generalize across domains.
> Guidance mechanism bridges domain gap: Our gradient-based constraint (Eq. 7) explicitly anchors generation to the medical domain through classical preprocessing.
> Scarcity of pre-trained medical diffusion models: Large-scale medical diffusion models don't exist at the scale/quality of Stable Diffusion.
> Application to color medical imaging modalities: Many modalities (histology, fundus, endoscopy, dermatology) are RGB, naturally suited to ImageNet priors.
>
> W3:
> We note that we already include NIQE (Natural Image Quality Evaluator), a no-reference perceptual quality metric, in Tables 1-2. Metrics like PSNR and SSIM can reward over-smoothing and blurred images may achieve high scores while destroying diagnostically critical edges. Our evaluation prioritizes task-based metrics because enhancement is a preprocessing step for diagnosis, not an end goal.
> We have added intensity distribution analysis to address this concern. We computed it for all three image types (degraded, classically enhanced, and our method) on the CAMUS test set. All values are in the [0,1] normalized range as required by the diffusion model:
> Mean Intensity: Degraded: 0.190 ± 0.036; Pseudo Label: 0.227 ± 0.024; Enhanced: 0.219 ± 0.024 Intensity Variability (Std): Degraded: 0.231 ± 0.034; Pseudo Label: 0.247 ± 0.020; Enhanced: 0.239 ± 0.020
> Our method introduces minimal intensity shift compared to classical preprocessing. We prioritize segmentation performance as our primary validation because perceptual quality metrics do not necessarily translate to clinical utility. While we have not conducted formal inter-observer studies, consistent improvements in boundary metrics (HD95, ASD) indicate enhanced anatomical delineation that translates to clinical measurements (ejection fraction, chamber volumes, wall thickness), making segmentation accuracy a direct proxy for diagnostic utility.
>
>
> W4:
> We performed further ablation to justify this (CAMUS Dice - U-Net):
> Classical pipeline alone: 0.839
> Diffusion without pseudo-label guidance: 0.484 (hallucinations).
> Diffusion with pseudo-label guidance (ours): 0.859
> Pseudo-labels provide noise-reduced targets for diffusion refinement, and differentiable gradients that classical methods alone cannot provide.
>
> W5:
> Synthetic degradation is the necessary and established approach in medical image enhancement research due to key constraints in acquiring paired medical data:
> Lack of paired real-world data:
> Paired clean/degraded images are clinically infeasible to acquire.
> Standard practices in medical imaging literatures:
> Low-Dose CT denoising: The widely-used Low-Dose CT Grand Challenge and AAPM datasets add synthetic Poisson noise to sinograms.
> MRI denoising: MRI denoising studies use BrainWeb images with synthetic Rician noise for controlled benchmarking.
> Physics-derived degradation models: Ultrasound speckle and MRI Rician noise are not arbitrary noise models; they're derived from underlying imaging physics and are widely used to simulate real-world degradations.
>
>
> Q1:
> Our reported results isolate enhancement quality by training on clean images. We now address the realistic scenario of training on degraded clinical data.
> Results (CAMUS U-Net):
> - Model trained on degraded → tested on ours: 0.862 (+0.003) from our result (0.859)
> - Model trained on degraded → tested on degraded: 0.841
> Enhancement benefits persist in realistic scenarios, validating genuine structural recovery rather than clean-data preprocessing artifacts, and confirming practical value for real-world clinical deployment.

---

### Author Rebuttal · Authors · 2026-01-25

We sincerely thank all reviewers for their thorough and constructive feedback. We are encouraged that reviewers recognize our work's novelty (R-L4s4, R-7Gxm), strong experimental validation (R-B8EW), and practical contribution to medical imaging (all reviewers). We have conducted additional experiments to address key concerns: (1) Supervised diffusion baseline showing our training-free method achieves comparable performance without training costs or overfitting risks, (2) Realistic scenario where segmentation models trained on degraded data still benefit from our enhancement, (3) Intensity distribution analysis confirming minimal intensity alteration while improving boundary accuracy.
We have carefully considered all feedback and have done our best to address the weaknesses identified by the reviewers. Below, we provide detailed responses to each concern.

---

### Meta-Review · Area_Chair_oUYo · 2026-02-03

**Recommendation:** Accept (Poster)
**Confidence:** 4

**Metareview:**

The limitation of using synthetic noise was pointed out by all reviewers, but L4s4 (weak accept) and B8EW (weak accept) seem satisfied that it doesn't disqualify the paper.  Reviewer 7Gxm maintained a borderline recommendation due to similar concerns of the synthetic noise not mapping well to real-world deployment, but overall, I think that, as L4s4 noted, this is OK for a methodology-focused conference paper. As such, I'm recommending a poster acceptance.

---

### Decision · Program_Chairs · 2026-02-13

Accept (Poster)